

# Satellite-based detection of deep convective clouds: the sensitivity of infrared methods, and implications for cloud climatology

Andrzej Z. Kotarba, Izabela Wojciechowska

Space Research Centre, Polish Academy of Sciences (CBK PAN), Bartycka 18A, 00-716 Warsaw, Poland

*Correspondence to*: Andrzej Z. Kotarba (akotarba@cbk.waw.pl)

**Abstract.** Reliable deep convective cloud (DCC) climatology relies heavily on accurate detection. Infrared-based algorithms play a critical role, as they are the only ones that can be applied to the 6.7 μm water vapour (WV) absorption band, and the 11 μm infrared (IR) window band. For over 40 years, the latter have been the only daytime/nighttime channels used in satellite cloud imaging. The study presents the first global validation of three, commonly-used DCC detection methods, which use
brightness temperature ($T_b$) in WV and IR bands. These methods are: the infrared window method (IRW; $T_{b11}$), the brightness temperature difference method (BTD; $T_{b6.7}-T_{b11}$), and the temperature difference method with the tropopause method (TROPO; $T_{b11}-T_{tropo}$). All methods were applied to one year (2007) of Moderate Resolution Imaging Spectroradiometer (MODIS) observations, and validated against collocated CloudSat-CALIPSO lidar-radar cloud classifications. Results indicate that even with optimal parameter configurations, DCC detection accuracy remains moderate, and below 75% (Cohen's $\kappa <$
0.4) for all methods. Global accuracy ranged from 56.6% (for TROPO) to 72.8% (for BTD) using an optimal threshold of −2 K. Regionally, the BTD method performs best, with accuracy of 72.9% over Europe, and 67.9% over Africa. Misclassifications are common with clouds such as Nimbostratus and Altostratus (single-layer cloud regimes) and Cirrus and Altostratus (multi-layer cloud regimes). Overall, the BTD method slightly outperforms the others, while TROPO is least effective. Our study highlighted the high sensitivity of these methods to threshold selection. Even a ±1 K change in the threshold resulted in a 10–
40% variance in DCC frequency. The latter finding is of particular importance for the construction of homogenous DCC datasets, whether as global mosaics, or as time series spanning multiple generations of satellite instruments.

## 1 Introduction

Deep convective clouds (DCCs) are formed through moist convection in the troposphere. DCC cloud top pressures may exceed ~450–500 hPa, and clouds may reach the tropopause or even penetrate the lower stratosphere. Although the least-frequent
cloud type on Earth (Sassen and Wang, 2008), they are the focus of scientific concerns due to their role in the hydrological cycle (Nesbitt et al., 2006), atmospheric chemistry (Wang and Prinn, 2000), and their association with severe weather events that include heavy precipitation, damaging wind, hail, tornadoes, or downburst phenomena (Taszarek et al., 2020). According to the European Environment Agency economic losses related to extreme climate events amounted to 738 billion EUR in EU Member States, and one third of that was caused by severe storms.



As the global climate warms, and more energy is being held in the atmosphere, troposphere dynamics are changing. In the mid-latitudes, the Hadley circulation is weakening and expanding poleward (Ceppi and Hartmann, 2016; Lu et al., 2007), causing change in the track and intensity of extratropical storms (Baatsen et al., 2015; Bender et al., 2012; Lehmann and Coumou, 2015). Consequently, convective processes are expected to intensify, and the frequency of DCC-related severe weather events may also increase (Aumann et al., 2008; Berthou et al., 2022). Identifying climate trends requires DCC time
series that span at least three decades, and a reliable reporting method.

The traditional (non-instrumental) approach to reporting is to observe the state of weather manually (visually), and to report DCC-related phenomena such as Cumulonimbus, hail, lightning, thunder, etc. (Taszarek et al., 2019). However, the limits of human perception make the method subjective and inaccurate, and limit spatial coverage. Additionally, it is only useful when a meteorological station is operated by a human (Eastman and Warren, 2014). Alternative techniques rely on ground- or sat-
ellite-based remote sensing. An orbital perspective is especially important for efficient mapping of DCCs, notably through the use of imagers that provide frequent, global coverage.

Passive satellite imagers detect DCCs based on their radiative properties. In the thermal infrared window (8–14 μm), DCCs are among the coldest objects in the field of view. Consequently, brightness temperature ($T_b$) thresholds can be set to discriminate between DCCs and the warmer background (Doelling et al., 2004; Gong et al., 2018; Govaerts et al., 2018; Mu et al.,
2017). However, the most important shortcoming of this method is that both DCCs and Cirrus are characterized by low $T_b$ thresholds in the infrared window. As a result, detection can be ambiguous; for example, Cirrus anvils associated with DCCs can be misclassified as convective clouds.

The $T_b$ threshold can be also applied to the water vapour absorption band (5.5–8 μm). In these wavelengths, electromagnetic emissions leaving Earth are absorbed by water vapour in the atmosphere as the signal propagates upward, toward space. As a
consequence, most radiance that is detected by a satellite originates in the upper atmosphere, including the highly-elevated tops of DCCs (Ackerman, 1996; Ai et al., 2017). However, difficulties are similar to those that arise with the thermal infrared window approach. Here too, Cirrus pose a challenge.

DCCs can also be detected using shortwave radiation (~4 μm or less), a combination of shortwave and longwave bands, or geophysical parameters retrieved from spectral radiances. One of the most widely adopted approaches is the algorithm imple-
mented by the International Satellite Cloud Climatology Project (ISCCP; Rossow and Schiffer, 1999). The ISCCP characterizes clouds based on their optical depth (*COT*; cloud optical thickness), and the atmospheric pressure at the cloud top (*CTP*; cloud top pressure) – measures are based on 10.5 μm brightness temperature, 0.65 μm reflectance, and radiative transfer modelling. As *COT* and *CTP* are continuous values, thresholds are applied to divide *COT-CTP* distributions into discrete classes. DCCs are identified when $COT > 23$ and $CTP < 440$ hPa.

A key shortcoming of the ISCCP classification, and other algorithms that exploit shortwave radiation, is that they are limited to daytime conditions. Tracking the diurnal DCC cycle requires a method that relies solely on longwave infrared observations. The design of such an algorithm is closely linked to the technical specification of the cloud imaging instrument. Most first-generation imaging sensors implemented as few as three spectral bands, two of which were dedicated to the infrared domain



(Holmlund et al., 2021). The infrared (IR) window channel, and the water vapour (WV) absorption channel were typically
centred at 11 μm, and 6.5 μm respectively. Advances in sensor technology have since made it possible to consider additional
IR spectral channels, such as the ozone thermal absorption band at 9.7 μm (Jurkovic et al., 2015). However, the long history
of 11 μm and 6.5 μm bands makes them indispensable for climatology, as they are the only way to derive multi-decadal DCC
time series.

One of the main disadvantages of IR-based approaches is the need for a threshold: the $T_b$ (or the $T_b$ difference) are used to
discriminate between DCCs and non-DCCs. Historically, thresholds were set arbitrarily, rather than being derived from an
empirical examination (see Sec. 2.2 for details). Notably, DCC detections were not validated, and accuracy assessments were
not reported (e.g. a classical confusion matrix for a binary classifier). The resulting DCC climatologies were only compared
with other (independent) datasets to check for discrepancies among sources (Sarkar et al., 2022; Sassen and Wang, 2008).
Such cross-comparisons cannot be considered as a substitute for validation.

The primary reason for the lack of validation was the absence of reliable 'ground truth'. In cloud research there is no 100%
accurate dataset, and any validation is in fact a relative comparison assuming that one source of observations is – for well
justified reason – more reliable than the other dataset (i.e. the one being validated). Currently, the state-of-the-art data for
validating cloud products originated from the CloudSat and Cloud-Aerosol Lidar and Infrared Pathfinder Satellite Observa-
tions (CALIPSO) missions. CloudSat hosted a cloud profiling radar, and CALIPSO hosted a cloud profiling lidar. Rather than
imaging the horizontal distribution of cloud, lidar and radar provide a vertically-resolved structure of the atmosphere. Due to
this unique capability, clouds are classified both during the day and at night based on their horizontal extent, height, thickness,
homogeneity, and presence of precipitation, rather than column-integrated radiances (Stephens et al., 2002; Winker et al.,
2003). An important consideration is that CloudSat and CALIPSO were configured to fly in close formation with the Aqua
satellite (Vincent and Salcedo, 2003), enabling quasi-simultaneous observation of clouds by lidar, radar, and Aqua's Moderate
Resolution Spectroradiometer (MODIS) instrument (Barnes et al., 1998).

The CloudSat-CALIPSO cloud typing algorithm was introduced by Wang and Sassen (2001), and its accuracy has been demon-
strated with surface-based lidar and radar observations. It was initially validated against visual (manual) cloud genera obser-
vations, performed at the lidar-radar location, and in accordance with World Meteorological Organization standards. The val-
idation study consisted of 540 cases, of which only four (according to reference data) or nine (according to the lidar-radar
classification) were DCCs. Wang and Sassen (2001) stated that the overall accuracy of their classification was 70%, but pro-
vided no specific details for DCC.

Sassen and Wang (2008) ran a post-launch assessment of the classification. The latter authors focused on one full year of
CloudSat observations (CALIPSO was excluded). Rather than performing a typical accuracy assessment, they only compared
zonally-averaged frequencies of individual cloud types. They found that the radar classification reported fewer DCCs than
ISCCP or surface-based data, and more As and Ns than the remaining databases. A similar study by Sarkar et al. (2022) noted
that difference between CloudSat-CALIPSO, ISCCP and surface-based visual observations was highest for DCCs. The latter
authors hypothesized that it may be related to the fuzzy logic used in the lidar-radar classification algorithm. Further validation





studies of CloudSat data have considered specific geophysical parameters, notably cloud base height (Candlish et al., 2013), precipitation (Kodamana and Fletcher, 2021), or cloud phase (Wang et al., 2024), but not the cloud classification.

Despite limitations, CloudSat and CALIPSO data, especially when combined into one, joint products, have been tested and adopted for validation purposes, including DCCs. Specifically, Yang et al. (2023) successfully demonstrated the potential of combining MODIS, CloudSat and CALIPSO data to validate IR-based DCC detection methods. However, their work only focused on the tropics (±25 °N), where DCCs and Nimbostratus were merged into a single category. Furthermore, their main objective was to establish a benchmark for their machine learning approach to DCC detection. Consequently, we still do not

know how accurate traditional, IR-based DCC detection methods are on a global scale. Are current thresholds appropriate, and do they guarantee optimal DCC detection accuracy? How sensitive is a DCC climatology to the selected threshold?

Given the importance of IR and WV heritage bands in long-term climatology, we perform the first, comprehensive, global-scale validation of critical IR-based DCC detection methods that rely on state-of-the-art CloudSat-CALIPSO lidar-radar cloud observations. Our overall question is: how consistent are DCC climatologies that are based on different IR methods, and

different DCC detection thresholds?

It is important to remember, that any time we use the term 'validation' refereeing to DCC detection methods and CloudSat-CALIPSO observations, we always mean a relative comparison between those datasets assuming lidar-radar data to be more accurate (active remote sensing methods, joint optical and microwave observations).

## 2 Data and methods

### 2.1 Database of collocated observations

Our validation of IR-based DCC detection methods required us to develop a dedicated database. Data consisted of temporally and spatially collocated observations of clouds performed with MODIS (Aqua), CloudSat and CALIPSO instruments. The specific data products we used were:

- 2B-CLDCLASS-lidar, version P1_R05; data are the result of a joint analysis of lidar (CALIPSO) and radar (CloudSat)
profiles, and provide information on cloud type. CALIPSO's lidar sampled the atmosphere at two wavelengths (532 and 1064 nm), every 333 m along the ground track, with a 90 m diameter footprint. CloudSat's radar operated similarly, but at a much longer wavelength: ~3190 nm (94 GHz), and with a noticeably larger footprint: 1.1×1.4 km. The two instruments were complementary: radar impulses can penetrate most cloud layers, but miss optically thin clouds, while the lidar signal is quickly attenuated, but is very sensitive to Cirriform. The 2B-CLDCLASS-lidar is designed
to take advantage of both systems, merging separate lidar and radar data into a single profile. The classification algorithm is run on cloud clusters, rather than individual profiles. Hence, the first step is to identify a cluster: namely, a group of horizontally connected cloud layers with similar vertical extent. Next, each cluster is characterized with respect to its geometrical and geophysical parameters (e.g. top and base heights, phase, temperature, maximum radar reflectivity, the presence of precipitation). Results are passed to a combined rule-based and fuzzy logic classifier,



which assigns one of eight possibilities: Cumulus (Cu), Stratus (St), Stratocumulus (Sc), Altostratus (As), Altocumulus (Ac), Nimbostratus (Ns), 'deep convective cloud' (Cumulonimbus, Cb) and 'high clouds', which includes all Cirriform (Cirrus, Cirrostratus, Cirrocumulus). For a more detailed description of the algorithm, see Sassen et al. (2008) and Wang and Sassen (2001). The data structure of the 2B-CLDCLASS lidar product supports reporting of up to ten cloud layers within a single profile. In our analysis, if at least one 'deep convective cloud' label was found

within a profile, the whole profile was designated as 'DCC', and as 'no-DCC' if this was not the case;

–    MYD021KM version C061; products provide calibrated radiances registered in 20 reflective solar bands (0.4–2.2 μm), and 16 thermal emissive bands (3.6–14.3 μm). The instrument operated as a passive imager, circling Earth twice each day, with a 2330 km-wide swath (Barnes et al., 1998). At nadir, the spatial resolution of MODIS imagery ranges from 250 m/pixel to 1 km/pixel, although atmospheric data products are released at 1 km/pixel resolution, or coarser

(Platnick et al., 2003). In order to match MODIS with CloudSat-CALIPSO observations, we only used 1 km data, and only for the two spectral bands of interest: 6.535–6.895 μm (central wavelength 6.715 μm, MODIS band 27), and 10.780–11.280 μm (central wavelength 11.030 μm, MODIS band 31). From these, we were able to calculate brightness temperature ($T_b$): specifically, $T_{bWV}$ or $T_{b6.7}$ for the WV absorption band, and $T_{bIR}$ or $T_{b11}$ for the IR window band. Geolocation information, which was necessary to spatially match MODIS with 2B-CLDCLASS-lidar data was

obtained for auxiliary MODIS products, namely MYD03 geolocation fields.

We considered a full year (2007) of MODIS, and CloudSat-CALIPSO observations. The initial database consisted of 175,666,879 matchups. In order to maximize data consistency, all MODIS data were parallax-corrected following the method reported in Wang et al. (2011). We decided to narrow the sample by rejecting observations that were too warm to feature a DCC. Specifically, all observations with $T_{bIR}-T_{bWV} < -10$ K were rejected. The latter procedure was implemented by Yang et

al. (2023), although the latter authors used a stricter threshold of −5 K. This left 9,507,319 matchups, which were evaluated. Table 1 provides more details on the composition of the sample. Aqua, CloudSat and CALIPSO were three independent spacecraft, and passed over the same location sequentially: CALIPSO was 15 seconds behind CloudSat, and CloudSat was 60 seconds behind Aqua. In Section 5 we address the potential impact of the sampling regime on the results of the validation.

**2.2 DCC detection methods**

We assess the accuracy of the following three IR-based methods:

–    The Infrared Window (IRW) method. In principle, this method is very simple. The only requirement is to set a $T_b$ value that can discriminate between DCCs (assumed to be colder), and a warmer background (either a cloudy or cloud-free atmosphere). There is no universal $T_{bIR}$ threshold for DCC detection, and different values have been used. Examples include: 210 K (Aumann and Ruzmaikin, 2013), 225 K (Aumann et al., 2018), 230 K (Hendon and Wood-

berry, 1993; Tissier and Legras, 2016), 235 K (Wall et al., 2018), and up to 245 K (Kubar et al., 2007). This ambiguity in threshold selection is reflected in studies by Mapes and Houze (1993), and Hong et al. (2006), who decided to





adopt two values: $T_{bIR} < 235$ K for the detection of 'high clouds' in general, and $T_{bIR} < 208/210$ K exclusively for DCCs.

— The Brightness Temperature Difference (BTD) method. In earlier work, Ackerman (1996) observed that in some
regions, $T_b$ at 6.7 μm could be greater than at 11 μm. In the tropics and mid-latitudes the $T_{bWV}$–$T_{bIR}$ difference was explained by the presence of thick clouds, notably DCCs. In general, a difference greater than 0 K coincides with clouds of $T_{bIR} < 210/215$ K, and the difference is highest when clouds reach the tropopause (Kolat et al., 2013). Although Ackerman (1996) suggested that $T_{bWV}$–$T_{bIR}$ could be used to detect thermal inversion in the polar troposphere, the method has been widely used to map DCCs, including the detection of overshooting tops (Bedka et al., 2010;
Martin et al., 2008).

— The TROPO method. Convective clouds cool as their tops penetrate up through the troposphere, and eventually cloud top temperature matches that of the tropopause. Hence, the difference between $T_{bIR}$ and the tropopause temperature ($T_{tropo}$) has been suggested as a DCC detection method. Zou et al. (2021) used $T_b$ at 8.1 μm, and suggested that a feature could be considered a DCC when $T_{bIR}$–$T_{tropo} \leq 7$ K. A similar value (6 K) was adopted by da Silva Neto (2016),
who used $T_{bIR}$–$T_{tropo}$ simultaneously with a more conventional $T_{bWV}$–$T_{bIR}$ approach. Aumann and Ruzmaikin (2013) set a $T_{bIR}$–$T_{tropo}$ threshold of 2 K, but used a climatological $T_{tropo}$ value instead of an actual (meteorological) value. The application of the TROPO method relies on $T_{tropo}$ data being available. In this study, the parameter was obtained from the Reanalysis Tropopause Data Repository (Hoffmann and Spang, 2022). Specifically, we refer to the first tropopause, as defined by the World Meteorological Organization, identified based on the ERA5 reanalysis.

The ISCCP scheme for DCC detection only was included for comparison. In the latter scheme, the cloud type classification is based on cloud optical thickness (*COT*), and cloud top pressure (*CTP*). Like IRW, BTD and TROPO methods, the ISCCP requires thresholds – for *COT* and *CTP* – which are also somewhat arbitrary (Rossow and Schiffer, 1999). Hahn et al. (2001) demonstrated that the ISCCP classes follow the traditional classification (i.e. the one implemented by the World Metrological Organization), but less strictly. Under the ISCCP paradigm, DCC detection requires *COT* and *CTP* information. The latter
were obtained from the MODIS MYD06 standard product (Platnick et al., 2003), with geometry and coverage that are identical to MYD21KM and MYD03 products. ISCCP results only refer to daytime conditions, while IRW, BTD and TROPO results are combined for day and night.

## 2.3 Measures of accuracy

The joint CloudSat-CALIPSO cloud classification was used as a reference. MODIS WV and IR $T_b$, and *COT* and *CTP* data
were used in IRW, BTD, TROPO and ISCCP methods to detect DCC. Agreement between the reference and validated methods was assessed based on a confusion matrix and related measures.

A confusion matrix is used to evaluate the performance of a binary classifier. It considers four possibilities: true positive, and true negative detection, when a method and the reference agree on the presence and absence of DCCs respectively; false positive detection, when a method finds DCCs, but the reference does not; and false negative detection, when a method reports





no DCCs but the reference does. The performance of a DCC detection algorithm can then be assessed with respect to: its overall accuracy; the probability of DCC detection ($PoD$); the DCC false discovery rate ($FAR$); and Cohen's Kappa coefficient ($\kappa$; Cohen, 1960):

$$\text{Accuracy} = \frac{TP+TN}{TP+TN+FP+FN} \, , \tag{1}$$

$$PoD = \frac{TP}{TP+FN} \, , \tag{2}$$


$$FAR = \frac{FP}{FP+TP} \, , \tag{3}$$

$$\kappa = \frac{2 \cdot (TP \cdot TN - FP \cdot FN)}{(TP+FN) \cdot (FN+TN) + (TP+FP) \cdot (FP+TN)} \, , \tag{4}$$

where: TP, TN, FP, FN are total number (counts) of true positive, true negative, false positive, and false negative detections respectively.

Values for a perfect classifier would be: accuracy and $PoD$ equal to 100%; $FAR$ as low as 0%; and $\kappa$ approaching 1.0 ($\kappa$ ap-

proaching 0.0 suggests that agreement between datasets was only achieved by chance, regardless of the actual accuracy).

DCC only made up 7% of CloudSat-CALIPSO observations investigated in this study, meaning that the sample was significantly unbalanced. It is reasonable to assume that the resulting accuracy measures are biased by the frequency of negative detections, which are far more likely than positive detections. To avoid this, we implemented bootstrap sampling (DiCiccio and Efron, 1996; Efron, 1979). First, the number of DCC in our CloudSat-CALIPSO reference dataset was determined. Then,

the reference was randomly sampled to identify exactly the same number of non-DCC observations. As a result, the count of DCC and non-DCC detections was equal. All accuracy measures were derived from this sub-sample. The procedure was then repeated 1000 times, and final accuracy statistics were calculated as the mean of the 1000 iterations.

## 3 Results

Rather than assuming a specific threshold for a DCC detection method, we tested a wide range of possible values (Fig. 1).

First, full statistics were derived for each instantaneous threshold, then we selected the threshold with the highest overall accuracy (or the highest $\kappa$ value, as both measures peaked at the same location). This value was then considered as the 'optimal' threshold, indicating that it guaranteed the best possible accuracy for a method.

All results were obtained  for land and ocean lying between 60 °N and 60 °S (the global domain), and for two smaller regions of interest: Europe (a mid-latitude, moist convection environment; 35 °N–60 °N, 15 °W–45 °E), and equatorial Africa (an

intertropical convergence zone, with very intense moist convection; 5 °N–15 °N, 20 °W–35 °E).

### 3.1 Highest achievable accuracy

The validation of IR-only DCC detection methods obtained  with the CloudSat-CALIPSO lidar-radar dataset showed that the highest achievable accuracy was moderate (Fig. 1). Depending on the method, and on a global scale, it varied between 56.6% (TROPO with a 15 K threshold) and 72.9% (BTD with a −2 K threshold). Regionally, accuracies were between 67.7% (IRW



with a 231 K threshold) and 72.9% (BTD with a −2.5 K threshold) for Europe, and from 65.6% (IRW with a 217 K threshold)
to 67.9% (BTD with a −1 K threshold) for Africa.

Our results showed that IRW and BTD methods performed almost equally well when global data were examined. Differences
in overall accuracy, detection probability, and the false alarm rate did not exceed ~5% at the global scale. However, changing
the spatial domain to Europe doubled discrepancies; the IRW method was less accurate, while the BTD method performed just

as efficiently as on the global scale. On the other hand, the comparison for tropical Africa found that both IRW and BTD
methods were less accurate.

Narrowing the spatial scale had the most significant impact on the performance of the TROPO approach. Using a single, global
threshold for the temperature difference between 10.8 μm and the tropopause proved impractical – the method detected DCC
with an overall accuracy of 56.6%, and a $\kappa$ coefficient of 0.13. At the regional scale, performance noticeably improved: DCC

detection probabilities doubled from 30% to 60%, resulting in a boost in overall accuracy of 14% in Europe, and 9% in tropical
Africa.

When IRW, BTD and TROPO methods were compared with the ISCCP daytime-only approach, the latter was found to be
more reliable in all respects. Not only did it outperform the other methods with respect to overall accuracy (76–77% regardless
of the spatial domain), but DCC detection probability was higher (80–85%), and, in general, the false alarm rate was lowest

among all of the investigated methods (25–28%).

### 3.2 Variability of thresholds

Inconsistency between global and regional results motivated us to test whether the optimal threshold for a method depended
on the latitude. We therefore derived accuracy measures for zones at 5 ° latitude, starting at the equator. With only one full
year of observations, it was impossible to obtain reliable results for latitudes above 40–50 °N/S (i.e. where DCCs are relatively

infrequent).

Our experiment revealed a clear relation between latitude and the optimal threshold. For the IRW method, DCC detection
across all latitudes was most accurate when the threshold was changed from ~218 K in the tropics to ~230 K in mid-latitudes.
The corresponding adjustment for the BTD method was only 2 K (from −1 K in the tropics to −3 K in mid-latitudes). Similarly,
for the TROPO method, the optimal threshold changed between low- and mid-latitudes. However, in this case, in the tropics

(20 °S–20 °N) it was constant (25 K), but dynamically decreased (to 10–15 K) at 40 °N/S.

Despite these adjustments in the threshold for different latitudes, the resulting DCC detection accuracy differed between zones.
Variability was greatest for the TROPO method, with ~15% amplitude for the overall accuracy index (Fig. 2g), and an even
more evident difference for the $\kappa$ coefficient (from 0.15 to 0.5; Fig. 2g). Additionally, highest and lowest scores differed most
over the northern hemisphere (30 °N and 50 °N). While this pattern was common to all methods, it was much less apparent

(by 50%) for IRW (Fig. 2e) and BTD (Fig. 2f) methods compared to TROPO.

Compared to IR-based approaches, the ISCCP DCC detection scheme performed almost equally well at all latitudes. DCC
detection probability remained high (80–90%), and the false alarm rate was relatively low (20–30%), resulting in a constant





overall accuracy of ~78–80% regardless of the latitude. Like IR-based methods, the ISCCP scheme performed better in northern mid-latitudes, but the improvement was small, and mostly seen in an increasing $\kappa$ coefficient (the consequence of a slight
increase in detection probability).

### 3.3 DCC misclassification

Accuracy below 100% necessarily indicates a certain degree of DCC misdetection by a given method. This could either be false negatives, when CloudSat-CALIPSO indicated a DCC but a method reported no DCC, or false positives, when reference data recorded no DCC, but a method reported one. In our study, false positives accounted for 8–17% of observations (mean
13%), while false negatives constituted 8–15% (mean 17%), depending on the method, and the spatial domain (global, Europe, or tropical Africa).

The lowest rate of false negative detections (8-10%) was found for the ISCCP method, but when only the IR method was considered, it was 15% at best for the BTD method, regardless of the sub-region of the study. Globally and over Europe, DCCs were most frequently missed by the TROPO method (35% and 19% of cases, respectively), while in Africa by the IRW method
(23% of cases). The better performance of the BTD method was at the price of a higher rate of false positive detections: the methods classified non-DCC as DCC in 12% (Europe, global) and 17% (Africa) of cases.

Since CloudSat-CALIPSO data label clouds at each level in the atmospheric profile, we were able to identify typical scenarios where a non-DCC observation was mislabelled as a DCC. We identified two underlying patterns. First, three types of clouds were most frequently (>90% of cases) misclassified as DCC: high clouds, As, and Ns. Second, which of these three types
dominated depended on whether we investigated a single- or multi-layer cloud scenario (Tab. 2).

IR-based methods classified As and Ns as DCC most frequently when clouds occurred as a single layer. This error accounted for over 35% of misclassifications for each cloud type – both globally, and for Europe. However, in tropical Africa, high clouds were more often mislabelled as DCC, although never as frequently as As or Ns (with the exception of the BTD method).

The situation changed significantly in a multi-layer cloud environment. In this scenario, we only focused on the top-most
(highest) cloud layer – the first to be detected when sensing from orbit. Our results showed that under such circumstances, IR methods falsely reported DCC when the atmospheric profile was topped with As or high (Cirrus-like) clouds. The co-occurrence of high clouds with other cloud types was the most challenging scenario, as this constituted up to 66% (TRW, TROPO) or 84% (BTD) of false positive detections. In the multilayer environment, Ns were not a problem; they accounted for 0.4% of erroneous cases, regardless of the method or the region.

It is important to note that misclassified As and Ns were the coldest clouds. The initial database of CloudSat/CALIPSO-MODIS matchups was screened for observations that satisfied the condition $T_{bWV} - T_{bIR} > -10$ K. Hence, all warm clouds, including a large share of As/Ns, were automatically excluded from further analysis. In the initial database of 175 million lidar-radar profiles, 50% of observations had $T_{bIR}$ in the range 250–282 K. After filtering (9.5 million profiles), the range shifted towards a noticeably colder regime, spanning 225–240 K.





## 3.4 Sensitivity to the selection of a threshold

Once we had calculated the optimal threshold for each DCC detection method, we then calculated mean seasonal DCC frequency for June-July-August 2005. The thresholds identified in the present study were applied to an independent dataset, namely geostationary Meteosat Second Generation data, which is collected every full hour. We used the 'High Rate Level 1.5 Image Data' product, based on Spinning Enhanced Visible and InfraRed Imager (SEVIRI) observations in two heritage bands: 6.25 μm and 10.8 μm (Holmlund et al., 2021). Data were accessed from the EUMETSAT archive (https://data.eumetsat.int/). The decision to use Meteosat data limited spatial coverage to a hemisphere; more specifically, to locations lying below SEVIRI's zenith angle (70 °) with respect to the sub-satellite point at 0 °E (hereinafter 'full disc'). Definitions of 'Europe' and 'tropical Africa' remained unchanged.

A sensitivity study provided two sets of statistics. In the first, we adopted a fixed threshold (Fig. 3 a–c), while in the second we used thresholds developed for each 5 ° latitude zone (Fig. 3 d–f). As the results of testing different latitudes (Sec. 3.2) only covered one year of observations, the transition in threshold values between zones was not smooth (Fig. 2 a–c), impacting the spatial distribution of DCC (Fig. 3 d–f). It is likely that artefacts could be eliminated with more data, and that the change in threshold could be continuous rather than incremental – however, both of these refinements were beyond the scope of this study.

DCC detection frequencies for the two approaches differed substantially (Tab. 3), demonstrating that the threshold selection significantly impacted the resulting DCC climatology. In the most extreme cases (the IRW method for Europe, and the TROPO method for Africa), latitude-adjusted thresholds doubled the frequency of DCC occurrence, compared to the fixed threshold approach. Although overall the change was lower, DCC frequency still increased or decreased by as much as one third. There were only two exceptions. When the BTD method was adopted globally, and the TROPO method was applied to Europe, the impact of the threshold selection was minor (a relative change of ~10% in mean DCC frequency).

Fig. 3 g–i shows that the tropics were most sensitive to threshold selection. Changing from a fixed, global threshold to a regionally-adjusted threshold reduced DCC frequency along the Intertropical Convergence Zone (ITCZ) when using IRW or BTD methods (by one third in relative terms), but increased DCC frequency for the TROPO method (rates at least doubled). The ITCZ was a dominant feature on DCC frequency maps—but only when the BTD method was used (Fig. 3b, e). ITCZ-comparable (or even higher) DCC frequencies were noted around Antarctica (IRW, TROPO; Fig. 3g, i), and over mountainous regions of Europe (the Alps, the Carpathians). The latter finding was particularly apparent when using the TROPO method with a fixed global threshold (Fig. 3c), and for the IRW method with a regionally-adjusted threshold (Fig. 3d).

To explore the sensitivity of DCC climatology to threshold selection in more detail, we calculated DCC frequency as a function of the threshold value (Fig. 4). The slope of the DCC frequency curve is the most important information to consider when testing sensitivity: a steep slope indicates a relatively large change in DCC frequency for a small change in the threshold value. We observed that for IRW (Fig. 4a) and BTD (Fig. 4b) methods the tropics (Africa) stood out as most sensitive to the choice of threshold. A shift of ±1–2 K in the threshold resulted in a change in DCC frequency that was two times larger than the



corresponding change over Europe, or at the global (full disc) scale. However, the same was not true for the TROPO method. This was due to an increase in sensitivity observed for Europe and the full disc (Fig. 4c): while the slope of the sensitivity

curve for both of these regions remained low when using IRW and BTD methods, it became as steep as the one noted for the tropics when using TROPO method. Consequently, the TROPO method was identified as most sensitive to the threshold value – regardless of the region it was applied to. On the other hand, the BTD approach was least sensitive to a change (the most desirable result), except for the tropics.

DCCs are very infrequent phenomena. Their frequency of occurrence is <0.1, on a scale where 0.0 means no DCC at all, and

1.0 indicates their permanent presence. A small change in that frequency (in terms of percentage points) translates into a high relative change. Here, we define relative change as the difference between the new value and the reference value, divided by the reference value. For data presented in Table 4, the reference was full disc DCC frequency with a fixed threshold, while the new DCC value was calculated using a threshold increased (or decreased) by 1 K.

Table 4 reveals that a change in the threshold of as little as 1 K can substantially affect the final climatological estimate of

DCC frequency. In the case of the BTD method, a ±1 K shift led to a ~40% relative change in DCC frequency, meaning that the frequency increased or decreased by nearly half of their absolute value (~0.01 for Europe and globally). An equally significant relative change was found for full disc and Europe, while Africa was close behind (~20% of relative change, 0.01 of absolute value). Comparisons of the other methods revealed relative differences in DCC frequency of 4–19%, typically values were close to ~10%.

**5 Discussion**

**5.1 Misclassification of DCC**

In this study, we evaluated three IR methods that are widely used to detect DCCs. We found that even when the optimal configuration is adopted, either globally or regionally, the final accuracy of the algorithm is only moderate (up to 73%). One factor that may have impacted our results is the reliability of reference data, namely the accuracy of the CloudSat-CALIPSO

cloud classification product (2B-CLDCLASS-LIDAR).

Assessments of CloudSat-CALIPSO cloud typing algorithm revealed partial disagreement in DCC frequencies between lidar-radar and other datasets, namely satellite-based ISCCP climatology, and surface-based visual (manual) classification (Wang and Sassen, 2001; Sassen and Wang, 2008; Sarkar et al., 2022). Specifically As and Ns clouds tended to be reported more frequently in CloudSat data. Such 'overrepresentation' could explain why these two cloud types were also the most frequently

considered to be DCC by the IR method, but as non-DCC by the lidar-radar reference. Possibly some As and Ns were actually DCC, and hence should be considered 'DCC' in CloudSat-CALIPSO reference used in this study. Such a procedure was implemented by Yang et al. (2023), who validated IR-based DCC detection methods in the tropics (25 °S–25 °N). The latter authors decided to merge Ns and DCCs into one category, and use it as a reference for DCCs. To test how such a strategy impacts detection accuracy, we repeated the Yang et al. (2023) study with two variants: with and without Ns in the reference.



The results showed (Tab. 5) that final accuracies were 4–5% higher when Ns was omitted, but did not differ significantly. We conclude that not combining Ns with DCC in our study had little impact on the final results.

In the multi-layer scenario, cloud misclassification was frequently the result of a method classifying high clouds (Cirrus) as DCCs. This can be explained by the simple fact that IR-based methods rely on cloud top temperature. Cirrus tend to be as cold as DCCs, and they can only be distinguished by examining their vertical extent or optical thickness. These parameters, however, are unavailable when IR and WV channels are considered. On the other hand, high clouds were also mislabelled as DCC in many ISCCP observations. ISCCP data include *COT*, therefore it is reasonable to expect fewer misclassifications. Unfortunately, ISCCP *COT* data are column-integrated, and Cirrus optical thickness is included in the optical thickness of underlying cloud layers (including Ns and As).

Limitations inherent in all types of cloud data make it impossible to develop a reference dataset that is 100% correct. Although CloudSat-CALIPSO is widely considered to be the most reliable current option, it is not free from its own misclassification issues. Importantly, all of the IR-based methods we assessed in this study were validated against exactly the same (common) reference. Therefore, if even the reference dataset has limitations, and actual (absolute) DCC detection accuracies may differ from those reported, relative differences between methods were captured correctly, and we were able to indicate which of them performed more or less efficiently.

## 5.2 Impact of matching geometry

A second factor that may have influenced the results of our study is the spatial and temporal collocation of lidar, radar, and MODIS observations. Each instrument was installed on a different satellite, meaning that a vertical atmospheric column was not observed simultaneously by all three sensors. In 2007, CloudSat (the Cloud Profiling Radar instrument) preceded CALIPSO (the Cloud-Aerosol Lidar with Orthogonal Polarization instrument) by ~15 seconds, and followed Aqua (MODIS) by approximately one minute.

The horizontal speed of a storm cloud is ~30–100 km h$^{-1}$, and a cloud could have shifted by ~0.5–1.5 km during the minute that separated MODIS and CloudSat-CALIPSO passes. This distance is generally within CloudSat's footprint (1.4 × 1.1 km), meaning that misclassification would only occur if CloudSat's ground track was collocated with the cloud's edge, and that the cloud moved outward relative to the ground track.

It should also be noted that a cloud is a 3-dimensional structure, which evolves vertically, especially when it is a DCC with a strong updraft. Updraft intensity varies from a few meters per second for fair weather cumuli (Kollias et al., 2001), to 10–30 m s$^{-1}$ for tropical cyclones (Stern and Bryan, 2018), and 30–50 m s$^{-1}$ for the most rapidly evolving DCCs (Apke et al., 2018; Musil et al., 1991). Therefore, cloud top height can increase by between 500 m (updraft ~8 m s$^{-1}$) to 2 km (updraft ~30 m s$^{-1}$) over one minute, corresponding to a decrease in cloud top temperature of 3–12 K (assuming a rate of 6 K km$^{-1}$). This creates a situation where CloudSat-CALIPSO could have detected a DCC that was not yet detected as a DCC by MODIS: the imager observed a cloud a few kelvins before it became a DCC.





The aforementioned scenario would result in more false negatives, reducing the overall accuracy of IR-based methods, compared to a scenario in which all sensors operated in collocated mode. The latter could be achieved if lidar, radar, and imager instruments were installed on the same platform, which is the case for the Earth Clouds, Aerosol and Radiation Explorer

(EarthCARE) satellite. The satellite was launched in 2024, and is in the commissioning phase at the time of writing.

EarthCARE hosts not only lidar and radar instruments, but also a 7-channel multispectral imager that covers three IR bands: 8.8 µm, 10.8 µm, and 12.0 µm (Illingworth et al., 2015). The setup eliminates all uncertainties related to spatial and temporal mismatches in a DCC observation. Unfortunately, the imager does not operate in WV absorption bands, ruling out the use of the two-channel DCC detection method. This situation is similar to CALIPSO's imager, the Imaging Infrared Radiometer

(IIR), which operated in three bands (8.7 µm, 10.5 µm, 12.0 µm). However, none of these channels consider WV absorption, as the sensor was optimised for joint CALIOP-IIR retrievals of Cirrus microphysical parameters.

Importantly, EarthCARE's radar – unlike CloudSat's – is a Doppler instrument. It provides data on the vertical velocity of hydrometeors (cloud particles, rain) with an accuracy better than 1.3 m s$^{-1}$ (Wehr et al., 2023). This may improve discrimination between cloud types, and provide more accurate labelling of DCCs, not only for validation, but also for training various

machine learning models (Afzali Gorooh et al., 2020; Kaps et al., 2024). Nonetheless, the use of imagers that include WV absorption bands makes MODIS-CloudSat-CALIPSO joint observations unique, and the most suitable for the evaluation of DCC detection methods.

**5.3 Implications for cloud climatology**

Given the limitations of the MODIS-CloudSat-CALIPSO cloud observing system, and based on our results, we conclude that

the $T_{bWV}-T_{bIR}$ brightness difference (BTD) method is slightly more robust among the evaluated algorithms. Accuracy was highest, as was the $\kappa$ coefficient, indicating best agreement with CloudSat-CALIPSO reference data. On the other hand, the BTD method resulted in the lowest DCC frequency among all of the considered algorithms, which, in turn, impacted the method's sensitivity to the threshold selection. Importantly, we found that the optimal global threshold was −2 K (−1 K in the tropics, −3 K at mid-latitudes). These values differ from a typically threshold of 0 K. Since higher thresholds lower DCC

frequency, the adoption of a 0 K threshold may underestimate DCC frequency, compared to the CloudSat-CALIPSO dataset. The BTD method requires data from IR window and VW absorption channels, and it can be easily applied to all generations of meteorological geostationary satellites. It supports the development of long term (40+ years) DCC climatologies, and composite DCC maps generated with data from various geostationary platforms such as the NCEP/CPC Level 3 Merged Infrared Brightness Temperatures product, the NASA SatCORPS Global Cloud Composite product, or the GEO-ring composites envi-

sioned for the ISCCP-Next Generation project.

The WV absorption band is typically not included on imagers that are hosted on polar-orbiting platforms. Examples include: the Advanced Very High Resolution Radiometer (NOAA-6/19, MetOp), the Visible/Infrared Imager Radiometer Suite (SNPP, NOAA-20/21, JPSS-3/4), the Multichannel Visible Infrared Scanning Radiometer (Feng-Yun-1 series), the Visible and Infra-



Red Radiometer (Feng-Yun-3A/C), and the VIS/IR Imaging Radiometer (Meteor-M series). Although these instruments have
been used since the late 1970s or early 1980s, none feature a spectral band in the 6.5 μm WV absorption region.

Detecting DCCs without the WV band requires using the single-channel IRW method, or IR window data with auxiliary
information on the tropopause temperature (the TROPO method). In this case, and when locally-adjusted thresholds are used,
IRW and TROPO methods produce comparable results, both in terms of overall agreement with CloudSat-CALIPSO, and the
spatial distribution of DCCs. We found that the TROPO method performed slightly better than the IRW approach, but only
for Europe, and only by 3–4%. This finding indicates that the inclusion of tropopause data does not necessarily lead to a
significant improvement in DCC detection, at least when the analysis explores the difference between the tropopause temper-
ature and cloud $T_b$.

Importantly, both TROPO and IRW methods recorded unexpectedly high DCC frequency, mostly over the southern hemi-
sphere (in winter). In these regions, DCCs were as frequent as in the ITCZ, which is not confirmed by any other dataset (Norris,
1998; Sarkar et al., 2022). This result suggests that the evaluated IR-based algorithms perform poorly, potentially due to the
misclassification of cold Ns and As, as these cloud types occur most frequently at higher latitudes (Chen et al., 2000). In turn,
this means that IRW and TROPO methods may be of limited use during colder seasons in higher latitudes – however, this is
the region that is best sampled by polar-orbiting spacecraft, and it is where climate change may impact DCC frequency the
most.

Our Meteosat-based sensitivity study demonstrated that the selection of an appropriate threshold is crucial for deriving accurate
DCC frequencies. This finding is important in order to be able to construct DCC climatologies from radiance time series
originating from various sensors (different families of sensors, or different generations within a single family). A 1 K difference
in $T_b$ may be a consequence of a difference in the spectral response function of different sensors, or it could be the consequence
of the sensor's calibration (Gunshor et al., 2004). The same DCC, observed simultaneously by two instruments, may appear
in the final dataset as an object with different $T_b$. Hence, using a single threshold for both datasets will produce incoherent
DCC climatologies.

If we take the CloudSat-CALIPSO classification as a reference, our study reveals the limits of the most common IR-based
DCC detection methods. Specifically, we identified thresholds that result in the highest achievable accuracy. However, better
results can be possibly achieved with other methods. We also evaluated DCC statistics resulting from the ISSCP cloud typol-
ogy. DCC detections based on *COT* and *CTP* were shown to be more reliable than any of the IR-based methods. This finding
demonstrates that daytime DCC detection benefits from the availability of shortwave radiances.

More complex algorithms, such as machine learning, can be used to process IR-only data. For instance, Yang et al. (2023)
trained their algorithm on $T_b$, but also used a number of derivative measures that addressed local variation (minima, maxima,
gradients). They achieved 72% accuracy in DCC detection. Even higher accuracy (98%) was reported by Chen et al. (2023),
who considered texture information, along with a sequence of images in three IR bands (6.25 μm, 10.7 μm, and 12.0 μm).

Since DCC detection requires both daytime and nighttime data, IR observations from cloud imagers will remain a primary
source of information. This is especially true for long-term climate studies, as the WV absorption channel, and the IR window



channel have been the only two heritage bands available on geostationary satellites since the early 1980s. Machine learning techniques that are trained on more sophisticated datasets (e.g. EarthCARE) offer the potential for more reliable and homogenous DCC climatologies that go beyond the limits of classical IR-based algorithms.

## 5 Summary and Conclusions

Our study explored the consistency of DCC climatologies derived from three widely-used IR-based DCC detection algorithms, namely: (1) the IR window method (11 μm spectral channel), (2) the brightness temperature difference (BTD) between 6.7 μm and 11 μm, and (3) the temperature difference between the tropopause and the 11 μm channel (TROPO). These algorithms were applied to MODIS/Aqua radiances, and compared with the unique, state-of-the-art CloudSat-CALIPSO lidar-radar cloud classification dataset. We assumed CloudSat-CALIPSO as a reference ('ground truth'). However, it must be acknowledged that in cloud research there is no universally reliable data set, and in a strict sense our study is a comparison with lidar-radar, which we have considered to be the most reliable of currently available.

In total 9,507,319 observations for 2007 were analysed, marking the first global scale evaluation of these DCC detection methods. The two key conclusions from our study are:

– IR-based methods demonstrate moderate accuracy in DCC detection ($< 75\%$; $\kappa < 0.45$), and only when regionally or zonally adjusted thresholds are used. Fixed, globally-applied thresholds should be avoided. Detection ambiguity arises from the misclassification of DCCs as Ns and As (in single layer cloud scenarios), or Cirrus and As (in multi-layer cloud scenarios). We conclude that these disagreements are partially due to the method's simplicity, but also identified uncertainties in the CloudSat-CALIPSO cloud classification, and imperfections in the observation system (notably, the temporal misalignment between lidar, radar, and imager observations);

– The high sensitivity of IR-based methods to threshold selection undermines the homogeneity of resulting DCC climatologies. Our analysis demonstrates that shifting the threshold by as little as $\pm 1$ K leads to a change in mean seasonal DCC estimates by 0.002-0.010, what translates into a relative change by 4%-40%. This finding is of particular importance when combining IR data from different sensors, whether to construct global mosaics, or to produce time series of DCCs from various generations of an instrument.

Assuming CloudSat-CALIPSO data as ground truth, we conclude that if the WV absorption and IR window channel are available, the BTD method should be prioritized over IRW and TROPO methods (e.g. for geostationary satellites). If these channels are not available (as with most polar-orbiting platforms) the TROPO method may provide comparable results, but at the cost of including auxiliary data on tropopause temperature. The IRW method should be considered as a last resort for detecting DCCs.

As technology progresses, the launch of new cloud imagers with more spectral channels (e.g. Flexible Combined Imager, GeoXO Imager, METimage) enhances the accuracy of DCC detection. This improvement will be especially notable when new data sources are paired with new data processing technologies, such as machine learning. Nevertheless, the WV absorption



channel, and the IR window channel remain the most important spectral bands for long-term DCC climate studies, along with IR-based methods that utilize them.

**Funding information**: This research was funded by the National Science Centre of Poland. Grant no. UMO-2020/39/B/ST10/00850.


**Acknowledgement**: We gratefully acknowledge Polish high-performance computing infrastructure PLGrid (HPC Center: ACK Cyfronet AGH) for providing computer facilities and support within computational grant no. PLG/2024/017024.

**Authors contribution**: Conceptualization: AK; Methodology: AK; Software: AK, IW; Formal analysis: AK, IW; Investiga-
tion: AK; Resources: AK; Data Curation: AK, IW; Writing - Original Draft: AK; Writing – Review and Editing: AK, IW; Visualization: AK; Project administration: AK; Funding acquisition: AK.

**Competing interests**: The authors declare that they have no conflict of interest.

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





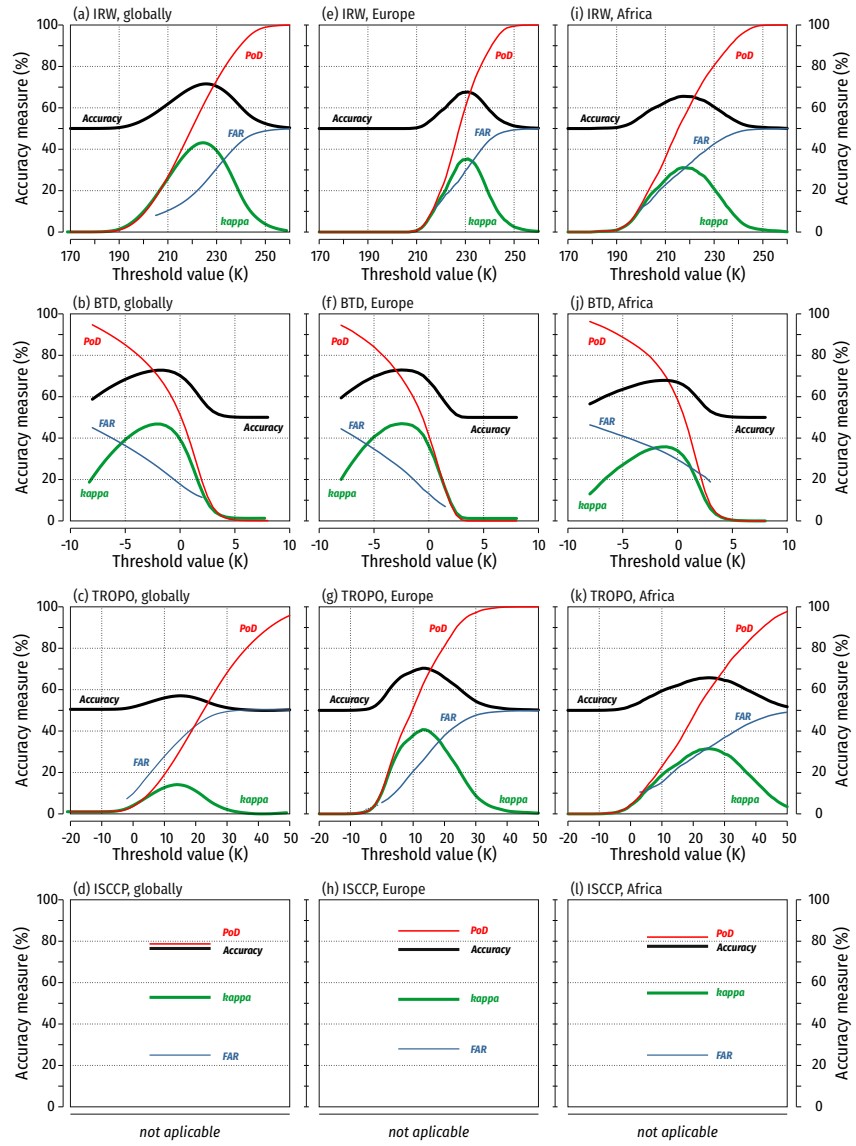

**Figure 1**. Accuracy of DCC detections using the infrared window (IRW) method, the WV-IR brightness temperature difference (BTD) method, the IR-tropopause temperature difference (TROPO) method, and the ISCCP method. Accuracy measures are shown as a function of the selected threshold (horizontal axis), except for ISCCP which uses a single set of parameter globally. Kappa (κ) values were multiplied by 100 to match the 0–100% range.





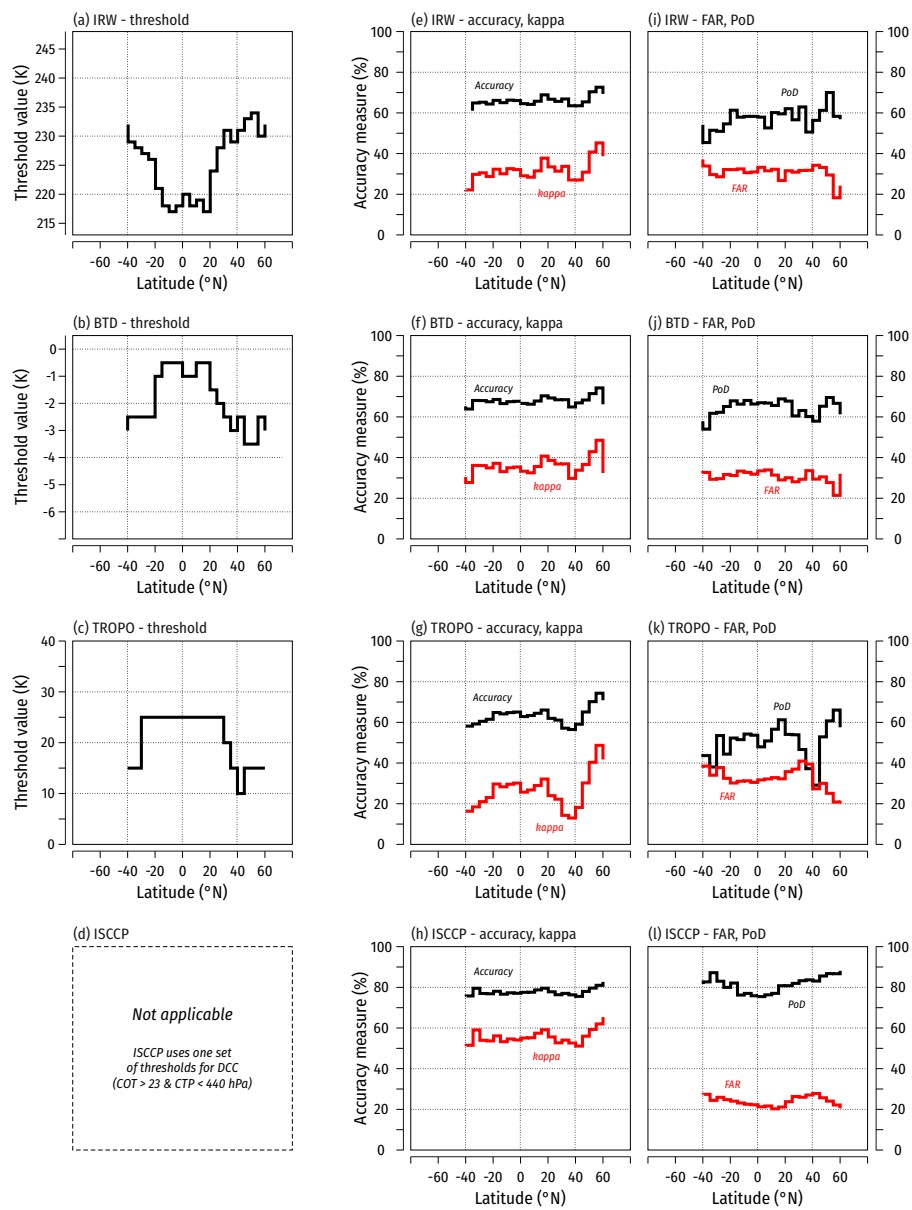

**Figure 2**. The optimal threshold for each method as a function of latitude (a–c), together with corresponding latitude-resolved measures of accuracy. Kappa (κ) values were multiplied by 100 to match the 0–100% range.



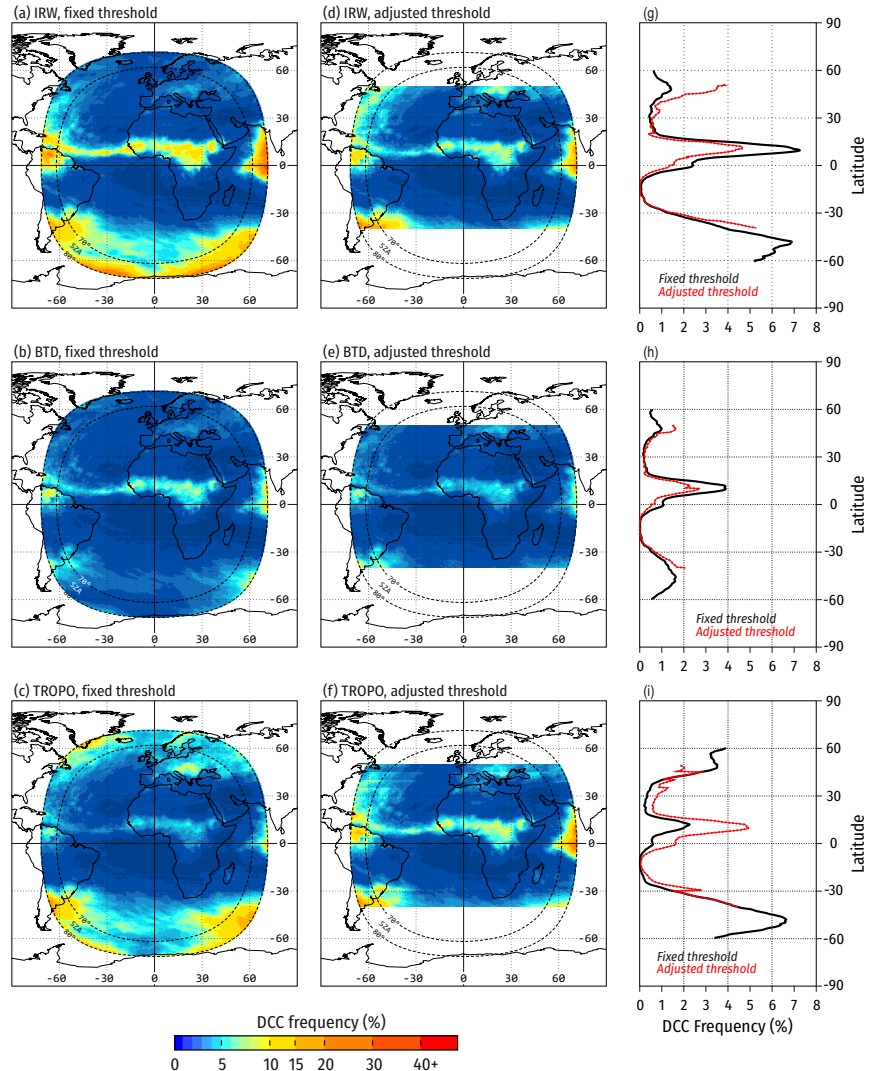

**Figure 3**. Mean seasonal (June-July-August 2005) DCC frequency, based on hourly Meteosat/SEVIRI data, and the methods evaluated in this study. Statistics were calculated using a fixed, global threshold (a–c), and latitude-adjusted thresholds (d–f); (d–i) summarize zonally-averaged DCC frequencies.



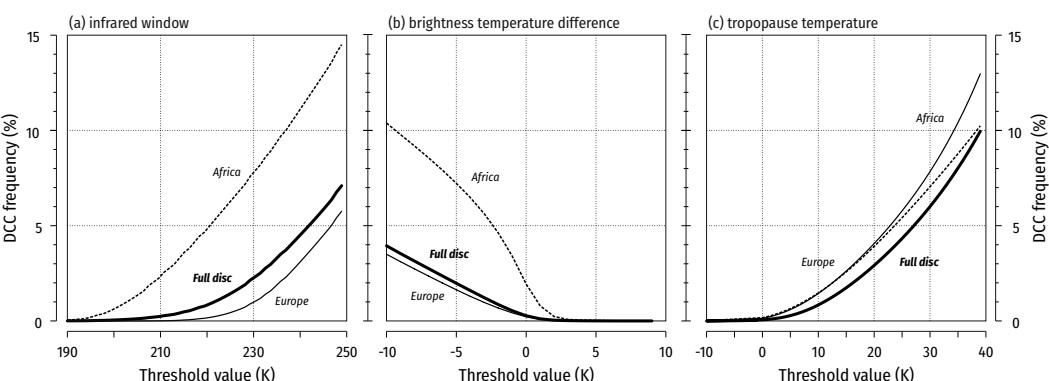

**Figure 4**. Mean seasonal (June-July-August 2005) DCC frequency as a function of the threshold applied to the DCC detection methods evaluated in this study.





1  **Tables**

2  **Table 1.** Cloud free (no DCC) and cloudy (DCC and other cloud present) percentages in the lidar-
3  radar profile data sample investigated in this study. Europe is defined as 35°N–60°N, 15°W–45°E, and
4  Africa is defined as 5°N–15°N, 20°W–35°E (see Sec. 3 for details on subregions).

| Region | Total number of observations | Cloud-free | Cloudy | | |
|--------|------------------------------|------------|--------|--|--|
| | | | no DCC | only DCC | not only DCC |
| Global | $n = 9{,}507{,}319$ | 3.7% | 89.2% | 7.1% | <0.1% |
| Europe | $n = 289{,}537$ | 2.5% | 94.3% | 3.2% | 0.0% |
| Africa | $n = 105{,}293$ | <0.1% | 85.0% | 14.9% | <0.1% |

6
7  **Table 2.** Percentage of cloud types classified by the four methods as DCC, but not reported as DCC in
8  CloudSat-CALIPSO observations. When more than one cloud layer occurred in a lidar-radar profile,
9  the cloud type refers to the highest layer (i.e. the first to be observed when looking from a satellite).

| | Cloud type frequency (%) (Single layer) | | | | Cloud type frequency (%) (Multilayer) | | | |
|--------|------|------|------|-------|------|------|------|-------|
| Method | High | As | Ns | Other | High | As | Ns | Other |
| *Global* | | | | | | | | |
| IRW | 16.6 | 44.4 | 38.8 | 0.2 | 64.6 | 35.2 | 0.1 | 0.1 |
| BTD | 8.8 | 37.8 | 40.8 | 12.6 | 59.7 | 36.8 | 0.2 | 3.3 |
| TROPO | 5.4 | 36.8 | 54.5 | 3.3 | 33.4 | 65.4 | 0.4 | 0.8 |
| ISCCP | 1.1 | 24.1 | 70.6 | 4.2 | 60.3 | 37.1 | 0.2 | 2.4 |
| *Europe* | | | | | | | | |
| IRW | 8.2 | 40.0 | 51.1 | 0.7 | 39.0 | 60.7 | 0.1 | 0.2 |
| BTD | 5.0 | 37.9 | 48.2 | 8.9 | 45.4 | 52.4 | 0.1 | 2.1 |
| TROPO | 5.7 | 35.8 | 56.4 | 2.1 | 29.6 | 69.9 | 0.2 | 0.3 |
| ISCCP | 0.7 | 21.7 | 72.9 | 4.7 | 43.4 | 54.9 | 0.2 | 1.5 |
| *Africa* | | | | | | | | |
| IRW | 17.6 | 52.1 | 30.4 | 0.0 | 65.9 | 33.8 | 0.1 | 0.2 |
| BTD | 29.1 | 50.7 | 17.5 | 2.7 | 83.5 | 16.2 | 0.0 | 0.3 |
| TROPO | 17.3 | 51.3 | 31.4 | 0.0 | 65.6 | 34.1 | 0.1 | 0.2 |
| ISCCP | 3.2 | 42.3 | 52.9 | 1.6 | 74.7 | 24.9 | 0.0 | 0.4 |

12  **Table 3.** Mean seasonal (June-July-August 2005) DCC frequency estimated using a fixed global
13  threshold, and a latitude-adjusted threshold.

| Detection method | DCC frequency with fixed (global) threshold | | | DCC frequency with latitude-adjusted thresholds | | |
|------------------|-----------|--------|--------|-----------|--------|--------|
| | Full disc | Europe | Africa | Full disc | Europe | Africa |
| IRW | 0.021 | 0.009 | 0.076 | 0.033 | 0.021 | 0.051 |
| BTD | 0.009 | 0.007 | 0.046 | 0.009 | 0.010 | 0.030 |
| TROPO | 0.017 | 0.026 | 0.026 | 0.028 | 0.023 | 0.054 |



**Table 4**. Mean seasonal (June-July-August 2005) DCC frequency calculated using a fixed global
threshold ('Reference DCC frequency'), and with thresholds increased and decreased by 1 K. Values
given in parentheses denote relative change, in other words, the difference between the reference DCC
frequency and the frequency after the threshold change, normalized with reference to the DCC
frequency.

| Detection method | Reference DCC frequency | Simulated DCC frequency with threshold changed by: | | | |
|---|---|---|---|---|---|
| | | −1 K | | +1 K | |
| *Full disc* | | | | | |
| IRW | 0.021 | 0.019 | (−11%) | 0.023 | (+8%) |
| BTD | 0.009 | 0.012 | (+42%) | 0.005 | (−37%) |
| TROPO | 0.017 | 0.015 | (−11%) | 0.020 | (+13%) |
| ISCCP | 0.027 | - | - | - | - |
| *Europe* | | | | | |
| IRW | 0.009 | 0.007 | (−19%) | 0.010 | (+14%) |
| BTD | 0.007 | 0.010 | (+43%) | 0.004 | (−38%) |
| TROPO | 0.026 | 0.024 | (-10%) | 0.029 | (+10%) |
| ISCCP | 0.034 | - | - | - | - |
| *Africa* | | | | | |
| IRW | 0.076 | 0.072 | (−5%) | 0.079 | (+4%) |
| BTD | 0.046 | 0.056 | (+22%) | 0.034 | (−26%) |
| TROPO | 0.026 | 0.023 | (−10%) | 0.028 | (+10%) |
| ISCCP | 0.075 | - | - | - | - |

**Table 5**. DCC detection accuracy in the tropics (±25 °N). DCC are defined as merged CloudSat-
CALIPSO DCC + Nimbostratus classes (as in Yang et al., 2023), and with DCC alone (as in this
study). The IRW method uses a threshold of 215 K, while the BTD method adopts a threshold of 0 K.

| Method | Ns+DCC as reference | | | | Only DCC as reference | | | |
|---|---|---|---|---|---|---|---|---|
| | Accuracy | PoD | FAR | κ | Accuracy | PoD | FAR | κ |
| IRW | 61.2% | 58.2% | 38.0% | 0.225 | 64.3% | 66.4% | 36.3% | 0.286 |
| BTD | 59.2% | 44.9% | 37.1% | 0.114 | 63.3% | 53.8% | 33.8% | 0.264 |



28 **Figures**

29

**Figure 1**. Accuracy of DCC detections using the infrared window (IRW) method, the WV-IR
brightness temperature difference (BTD) method, the IR-tropopause temperature difference (TROPO)
method, and the ISCCP method. Accuracy measures are shown as a function of the selected threshold
(horizontal axis), except for ISCCP which uses a single set of parameter globally. Kappa ($\kappa$) values
were multiplied by 100 to match the 0–100% range.

**Figure 2**. The optimal threshold for each method as a function of latitude (a–c), together with
corresponding latitude-resolved measures of accuracy. Kappa ($\kappa$) values were multiplied by 100 to
match the 0–100% range.

**Figure 3**. Mean seasonal (June-July-August 2005) DCC frequency, based on hourly Meteosat/SEVIRI
data, and the methods evaluated in this study. Statistics were calculated using a fixed, global threshold
(a–c), and latitude-adjusted thresholds (d–f); (d–i) summarize zonally-averaged DCC frequencies.

**Figure 4**. Mean seasonal (June-July-August 2005) DCC frequency as a function of the threshold
applied to the DCC detection methods evaluated in this study.