# Peer review of "Satellite-based detection of deep convective clouds: the sensitivity of infrared methods, and implications for cloud climatology"

_EGUsphere, 2024_

## Referee Comment (RC2)

**Evaluation**

The focus of this analysis was to evaluate the accuracies of commonly used infrared-based deep convective cloud (DCC) detection methods, with additional comparison to ISCCP scheme for cloud classification and detection. These methods were evaluated based on how well they detected CloudSat-CALIPSO identified DCCs. The main finding was that the brightness temperature difference method performed the best out of the three infrared-based detection methods, and that a fixed 11-micron IR threshold should not be used for detecting DCCs. This study is important given the robust literature on deep convective system climatologies from a passive remote sensing perspective that often use single 11-micron IR thresholds for distinguishing DCCs. Furthermore, the data sets used and accuracy tests done in this study seem to be thorough, and limitations were addressed in great detail. I recommend the manuscript for publication after addressing the comments below.

**Specific Comments**

*L149*: What was the reasoning to use the -10 K threshold as opposed to the stricter -5 K? Was it to ensure sufficient sampling? And was the -10 K threshold applied as a filter before seeing if CloudSat-CALIPSO flagged any DCCs?

*L211-212*: Could you please expand upon what exactly was repeated? Do you subsample the DCC sample as well as the non-DCC sample? Since you repeat it 1000 times, do each of your subsamples contain unique cases (i.e. they don't contain the same cases as the other iterations?)

*L234-236*: Interesting, why do you think the TROPO approach improved while the other methods were less accurate for a smaller domain?

*L243*—"derived accuracy measures for zones at 5° latitude": How did you do this? Did you calculate confusion matrices for a range of thresholds and then define which one was optimal? If so, how did you select the range of thresholds?

*L308-310*: Please clarify explanations, specifying what "change" and "increase" / "decrease" is referencing (i.e. what are you comparing to?). Also, the "two exceptions" are not clearly defined—exceptions to what?

*L315*—"Antarctica": I would suggest specifying that the DCC occurrence is over the Southern Ocean (not Antarctica). Also, do the authors think that high DCC frequency over this region is real? Or is the fixed threshold perhaps detecting another type of high cloud? It would be beneficial if the authors could comment on this point, especially if it is an issue with using a fixed threshold because it can misclassify DCCs.

*L316*—sentence beginning with "The latter finding": This sentence might be more impactful if it was rephrased to explain whether an increased DCC frequency over the mountainous regions is valid.

*L344-345*: Since this whole section does not discuss the accuracy of CloudSat-CALIPSO, perhaps restructure this paragraph and the next by joining the paragraphs. Also, what other inaccuracies were there other than the potential misclassification of Ns that was then stated not be a significant issue in L355?

*L348-349*: Are there references stating that CloudSat is overrepresenting As and Ns clouds? It would be particularly interesting to share those studies since CloudSat and CALIPSO are generally considered to have a better representation of clouds than ISCCP.

*L357-363*: Is it possible to quantify the significance of the inaccuracies of misclassifying clouds due to multi-layer scenarios in each of the methods, including in the ISCCP data? If not, do you have a qualitative sense of how significant the misclassification is (i.e. does misclassifying multi-layer cloud scenarios lead to the most inaccuracies in detecting DCCs)?

*L430-431*: It is unclear what the authors mean in this sentence. Are the authors suggesting that it is likely that DCCs occur as frequently in higher latitudes as they do in the ITCZ? Also, what is the "misclassification of cold Ns and As" referring to: the initial $Tb_{WV} - Tb_{IR} > -10$ K screening, or CloudSat-CALIPSO potentially overestimating As and Ns clouds?

*L469-470*: What specifically are the uncertainties in CloudSat-CALIPSO cloud classification, other than the As/Ns overrepresentation that was deemed insignificant in L355?

**Technical Comments**

*L111*: "referring" instead of "refeering"

*L113*: It is unclear what the text in parentheses is referring to

*L151-153*—sentence beginning with "Aqua…": I think this information should go at the beginning of the section before introducing the individual data products to give some context on the observations.

*L178-179*: what does "the first tropopause" mean?

*L246-247*: If I understand correctly, this statement can be generalized to say that overall accuracy increases by using multiple TB thresholds that change with latitude?

*L285*—sentence beginning with "It is important…": Perhaps rephrase as it is unclear what is meant by this sentence.

*L296*: omit "to a hemisphere; more specifically," as this explanation is not quite accurate.

*L340*: Discussion section is section 4 (not 5)

Figure captions: Perhaps specify what each row and column correspond to, as well as the colors of each line, to make it quicker to study each figure.

---

## Author Response (AR1)

Response to: **Reviewer #1**

Overall evaluation: In this manuscript, the authors carried out a series of validations and experiments on three different DCC classification algorithms against observations from CloudSat-CALIPSO lidar-radar observations. This research would be helpful for gaining insights on how the popular algorithms perform with different thresholds and over different regions. The major drawback is on the paper writing, especially in the introduction part which consists numerous unclear descriptions, including but not limited to my specific comments #1~12 below. Therefore, I would recommend a minor revision from the authors to make this manuscript better reader-friendly to the general peers in and outside the relevant field before it gets published.

Specific comments:

— Line 24, 'Although the least-frequent cloud type' should be 'Although/Despite being the least frequent cloud type'.

Corrected, as suggested.

— Line 28, there should be a comma between 'Agency' and 'economic'.

Corrected, as suggested.

— Line 30, 'As global climate warms' should be 'With global warming'.

Corrected, as suggested.

— Line 38, 'limit spatial coverage' should be 'the spatial coverage is limited'.

Corrected, as suggested.

— Line 38-39, I believe that 'only useful when operated by a human' can be part of 'limited spatial coverage', since it is aforementioned as 'manual observation'.

Rephrased, as suggested.

— Line 41, it should be like 'imagers that provide frequent observations with global coverage'.

Corrected, as suggested.

— Line 52, it should be 'Here, Cirrus pose a threat as well' to be more formal in writing.

Corrected, as suggested.

— Line 54, for 'spectral radiances', do you mean 'multispectral radiances'?

Yes, 'multispectral' sounds more appropriate.

— Line 57, 'measures' should be 'measurements'.

Corrected, as suggested.

— Line 75-76. This is a relatively big statement which needs some reference citing to support it, like how even observations from CALIPSO cannot be regarded as direct observation or 'ground truth'.

Unfortunately, this is true in cloud studies. There is no 100% reliable source of reference data for cloud detection and classification. The only solution is to use a reference dataset that is believed to be more accurate than others. That said, 'believe' is not a matter of guessing, or a random choice, but results from a deep understanding of the physics of (remote) sensing. For instance, active profiling with lidar is much more accurate for cloud detection than passive imaging in the infrared range, thus lidars (like CALIOP) have been widely used for validating imagers. A true 'ground truth' that reports clouds at every scale, at any time, and at every wavelength is the Holy Grail for cloud remote sensing. And we wish we could have it!

— Line 92, what do you mean by 'the latter authors', as there is only one group of authors mentioned in this newly-beginning paragraph.

Clarified, as suggested.

— Line 95, the full names of As and Ns should be presented.

Clarified, as suggested.

— Line 173, by mentioning Tb used by Zou et al., (2021), do you mean they are using observations from 8.1 μm as a symbol for $T_{tropo}$? It seems like the observation or derivation of $T_{tropo}$ is never mentioned in this paragraph.

Corrected. $T_b$ stands for $T_{bIR}$ (the 'IR' in the subscript was missing).

— Line 180, which scheme is the 'latter' one?

Clarified, as suggested.

— Line 293, by 'every full hour' do you mean data collected hourly?

Clarified, as suggested (hourly and every full hour)

— Line 296~297, I find this sentence extremely difficult to understand (or guess) the meaning.

Rephrased, as suggested.

— Line 299, what numbers are selected as the 'fixed thresholds' for each method?

Clarified, as requested. Threshold values have been added.

— Line 366~369. For the discussion, since the reference dataset can also have misclassified DCCs, would the relative differences between different methods be affected as well, such as one method happens to misclassify some DCCs in the same way as the reference when the other doesn't but give the correct results instead?

There is no reason or evidence to suspect that any of the evaluated methods miss some clouds more or less frequently than others, or the reference. These methods are 'genetically' similar,

all three rely on passive sensing in the thermal infrared range. Perhaps the use of another type of detection method (e.g. machine learning or object-based), would introduce the suggested inhomogeneity. But at this stage, and with the available data, we are not in a position to draw any conclusions on the matter.

The focus of this analysis was to evaluate the accuracies of commonly used infrared-based deep convective cloud (DCC) detection methods, with additional comparison to ISCCP scheme for cloud classification and detection. These methods were evaluated based on how well they detected CloudSat-CALIPSO identified DCCs. The main finding was that the brightness temperature difference method performed the best out of the three infrared-based detection methods, and that a fixed 11-micron IR threshold should not be used for detecting DCCs. This study is important given the robust literature on deep convective system climatologies from a passive remote sensing perspective that often use single 11-micron IR thresholds for distinguishing DCCs. Furthermore, the data sets used and accuracy tests done in this study seem to be thorough, and limitations were addressed in great detail. I recommend the manuscript for publication after addressing the comments below.

Specific Comments

— L149: What was the reasoning to use the -10 K threshold as opposed to the stricter -5 K? Was it to ensure sufficient sampling? And was the -10 K threshold applied as a filter before seeing if CloudSat-CALIPSO flagged any DCCs?

We have **added** a sentence to clarify our reasoning. The $-5K$ threshold used in the other study was only applied to the tropics, while we aimed for a global study. Therefore, we decided to adopt a more liberal threshold for DCC pre-selection, and allow more non-DCC clouds to be accounted for. In our opinion, this resulted in a more reliable estimation of accuracy measures, as more potential sources of misclassification were considered.

— L211-212: Could you please expand upon what exactly was repeated? Do you subsample the DCC sample as well as the non-DCC sample? Since you repeat it 1000 times, do each of your subsamples contain unique cases (i.e. they don't contain the same cases as the other iterations?)

This has been **clarified** in the manuscript. The DCC sample was always the same, only the non-DCC subsample changed (1000 times). There were 1000 iterations, and each returned instantaneous accuracy measures. In the final step, these 1000 instantaneous measures were averaged and returned a single statistic (the bootstrap estimation of a parameter). This single value is reported in the paper.

— L234-236: Interesting, why do you think the TROPO approach improved while the other methods were less accurate for a smaller domain?

Broadly speaking, the tropopause exists in two very distinctive 'modes': a tropical mode, and an extra-tropical mode. Although this is still the same tropopause, the tropical tropopause is much colder and much higher than the tropopause at higher latitudes. The transition between the two is sharp. Using a single threshold for the TROPO method for the whole globe does not capture this significant difference, and rapid change. Locally-adjusted thresholds do. The parameters considered in BTD and TRW methods do not show such large and sharp change with latitude.

— L243—"derived accuracy measures for zones at 5° latitude": How did you do this? Did you calculate confusion matrices for a range of thresholds and then define which one was optimal? If so, how did you select the range of thresholds?

An explanation has been **added** to the text. The reviewer is correct. The procedure for the selected 5-degree zones was exactly the same as for the entire globe, except that the input data was pre-filtered to one latitude. The threshold range was also the same as for the whole globe

calculation. The selection of the range of thresholds resulted from our initial trials. We started with a very wide range, including values that were 100% too high or too low, and then gradually narrowed the selection down to the range reported in the paper.

— L308-310: Please clarify explanations, specifying what "change" and "increase" / "decrease" is referencing (i.e. what are you comparing to?). Also, the "two exceptions" are not clearly defined—exceptions to what?

This has been **clarified** in the manuscript. The paragraph compares two approaches and explains any differences/changes between the two. Specifically, the latitude-adjusted approach is compared to the fixed-threshold approach.

— L315—"Antarctica": I would suggest specifying that the DCC occurrence is over the Southern Ocean (not Antarctica). Also, do the authors think that high DCC frequency over this region is real? Or is the fixed threshold perhaps detecting another type of high cloud? It would be beneficial if the authors could comment on this point, especially if it is an issue with using a fixed threshold because it can misclassify DCCs.

**Changed** and clarified in the manuscript.

— L316—sentence beginning with "The latter finding": This sentence might be more impactful if it was rephrased to explain whether an increased DCC frequency over the mountainous regions is valid.

At this point we do not have sufficient data to provide a conclusive explanation. It is known that mountain ridges favour the formation of convective clouds, including DCCs. All of the investigated methods captured the phenomena. However, the intensity of DCC production over mountains (i.e. the actual frequency of DCC) requires a more localised study, with precisely tailored reference data (e.g. locally-operated cloud radar). Our paper is part of a larger project in which we hope to address the issue of the impact of topography on DCC frequency in greater detail.

— L344-345: Since this whole section does not discuss the accuracy of CloudSat-CALIPSO, perhaps restructure this paragraph and the next by joining the paragraphs. Also, what other inaccuracies were there other than the potential misclassification of Ns that was then stated not be a significant issue in L355?

We have **added** a more detailed explanation of the identified discrepancies. We clearly state that the misclassification is most likely to be attributed to insufficient spectral information (only one or two spectral bands), which means that a simple DCC detection algorithm cannot accurately discriminate between DCC and spectrally similar clouds (including Cirrus or cold Ns).

Regarding the suggestion to merge the two first paragraphs, as the first one simply introduces the discussion, we prefer to keep it as a separate section of text.

— L348-349: Are there references stating that CloudSat is over representing As and Ns clouds? It would be particularly interesting to share those studies since CloudSat and CALIPSO are generally considered to have a better representation of clouds than ISCCP.

We did our best to find validation studies of CloudSat-CALIPSO cloud type data. The overestimation statement reported in the manuscript is our conclusion from Wang and Sassen (2001), Sassen and Wang (2008), and Sarkar et al. (2022) , which are all cited in our paper. We welcome

any suggestions regarding other validation studies/ reports. We believe that the upcoming observations of the EarthCARE mission may provide us with more accurate cloud type data from joint lidar-radar satellite observations, but they will not be applicable to CloudSat-CALIPSO, unfortunately.

— L357-363: Is it possible to quantify the significance of the inaccuracies of misclassifying clouds due to multi-layer scenarios in each of the methods, including in the ISCCP data? If not, do you have a qualitative sense of how significant the misclassification is (i.e. does misclassifying multilayer cloud scenarios lead to the most inaccuracies in detecting DCCs)?

This is a very interesting question. Yes, we can provide quantitative information. In general, there are two possible vectors of misclassification: commission error (a method reports a 'DCC', when CloudSat-CALIPSO reports 'no DCC'), and omission error (a method reports 'no DCC' while CloudSat-CALIPSO reports a 'DCC'). Table R1 gives detailed statistics on the frequency of both types of errors, stratified by the number of cloud layers found in the atmospheric column by CloudSat and CALIPSO.

**Tab. R1**. Percentage of observations (%; $n$=9,507,319) when either an error of omission (false negative DCC) or commission (false positive DCC) was detected. Results are given for three cloud co-occurrence scenarios: no clouds (0 layers; 4% of cases), clouds only in one layer (32% of cases), and multilayer clouds (2+ layers; 63% of cases). The number of layers is according to CloudSat-CALIPSO observations.

| Method | Commission error | | | Omission error | | | Errors (total) |
|--------|---------|---------|----------|---------|---------|----------|--------|
| | *0 layers* | *1 layer* | *2+ layers* | *0 layers* | *1 layer* | *2+ layers* | |
| IRW | 0.03 | 12.12 | 8.38 | 0.00 | 2.04 | 0.00 | 22.57 |
| BTD | 0.77 | 14.13 | 8.29 | 0.00 | 1.62 | 0.00 | 24.81 |
| TROPO | 0.15 | 11.41 | 3.71 | 0.00 | 4.46 | 0.00 | 19.73 |
| ISCCP | 0.00 | 7.24 | 4.43 | 0.00 | 4.27 | 0.00 | 15.94 |

Based on Table R1, we conclude that regardless of the method, the structure of errors was dominated by commission errors when only a single layer of clouds occurred.

We have added a paragraph to the manuscript to comment on this finding.

— L430-431: It is unclear what the authors mean in this sentence. Are the authors suggesting that it is likely that DCCs occur as frequently in higher latitudes as they do in the ITCZ? Also, what is the "misclassification of cold Ns and As" referring to: the initial $T_{bWV}-T_{bIR} > -10$ K screening, or CloudSat-CALIPSO potentially overestimating As and Ns clouds?

This has been rephrased for clarity. Similarities in the frequency of DCC in the ITCZ and at high latitudes were revealed by the discussed methods. We therefore conclude that this was due to incorrect information about DCC frequency, as the ITCZ features much more DCCs than any other region on the planet – as demonstrated by other studies (cited in the manuscript). The investigated methods may have reported too many DCCs at higher latitudes because they (erroneously) considered Ns and As with cold cloud tops as DCC. Therefore, the misclassification appears to be related to DCC detection performance, and not the CloudSat-CALIPSO reference.

— L469-470: What specifically are the uncertainties in CloudSat-CALIPSO cloud classification, other than the As/Ns overrepresentation that was deemed insignificant in L355?

L355 refers specifically to the approach of Yang et al. (2023), who used the same label, 'DCC', for two different cloud types: DCC and Ns (in the CloudSat-CAPISO reference).

We wanted to check whether or not the identification of Ns as DCC, and including them in a reference dataset, could have affected our results. Hence, we calculated statistics: 1) assuming that Ns were indeed DCC, and merging Ns and DCC into one class as in Young et al. (2023), and 2) assuming that Ns were a separate cloud type class, and not merged with DCC (as in our study). The differences were not significant. We believe that this is the 'insignificance' the reviewer has in mind. However, this difference only concerned the studied comparison, and not the CloudSat-CALIPSO reference itself.

Alternatively, the Reviewer may be referring to the uncertainty related to CloudSat-CALIPSO data in general. Again, it is very difficult to draw a conclusion, as discussed in the paper. In fact, we found this question to be the most challenging part of the study. The only investigation that validated the radar-based cloud type, in the form of a confusion matrix, was that of Wang and Sassen (2001; cited in the manuscript). The table below summarizes the findings. Note that not a single DCC was observed during the validation campaign! In the absence of direct validation, our discussion was limited to uncertainties in the cloud type classification based on CloudSat, CALIPSO, ISCCP, and ground-based observations reported in other studies, as stated in our paper.

TABLE 3. Contingency table showing comparison of cloud type derived from the algorithm vs cloud type from human visual observations for Jan 1998.

| | | Algorithm | | | | | | | | |
|---|---|---|---|---|---|---|---|---|---|---|
| | | High | As | Ac | St | Sc | Cu | Ns | Deep | Clear |
| Obs | High | 39 | 2 | 3 | 0 | 2 | 1 | 1 | 0 | 7 |
| | As | 10 | 37 | 1 | 0 | 0 | 0 | 0 | 0 | 0 |
| | Ac | 13 | 1 | 15 | 4 | 1 | 0 | 0 | 0 | 7 |
| | St | 0 | 1 | 0 | 55 | 12 | 0 | 12 | 0 | 1 |
| | Sc | 0 | 0 | 0 | 8 | 13 | 0 | 0 | 0 | 2 |
| | Cu | 0 | 0 | 0 | 0 | 0 | 0 | 0 | 0 | 0 |
| | Ns | 0 | 0 | 0 | 0 | 0 | 0 | 20 | 0 | 0 |
| | Deep | 0 | 0 | 0 | 0 | 0 | 0 | 0 | 0 | 0 |

Table from *Wang and Sassen* (2001), DOI: 10.1175/1520-0450(2001)040<1665:CTAMPR>2.0.CO;2

Technical Comments

— L111: "referring" instead of "refeering"

Corrected, as suggested.

— L113: It is unclear what the text in parentheses is referring to

Clarified, as requested.

— L151-153—sentence beginning with "Aqua…": I think this information should go at the beginning of the section before introducing the individual data products to give some context on the observations.

Changed, as suggested.

— L178-179: what does "the first tropopause" mean?

After Xian and Homeyer (ACP, 19, 5661–5678, 2019; 10.5194/acp-19-5661-2019): *The WMO definition defines the first tropopause in a profile as "the lowest level at which the lapse rate falls to 2 °C·km−1 or less, provided the average lapse rate between this level and all higher levels within 2 km also does not exceed 2 °C·km−1" (WMO, 1957). The WMO definition allows for a secondary tropopause "if above the first tropopause the average lapse rate between any level and all higher levels within 1 km exceeds 3 °C·km−1, then a second tropopause is defined by the same criterion."*

— L246-247: If I understand correctly, this statement can be generalized to say that overall accuracy increases by using multiple TB thresholds that change with latitude?

No, it means that the optimal threshold for a method (i.e. the one that results in the best accuracy for that method) is not the same for all locations on Earth; it varies with latitude. Fig. 2a–c shows how it varies, while Fig. 2e–l illustrates the corresponding accuracy.

— L285—sentence beginning with "It is important…": Perhaps rephrase as it is unclear what is meant by this sentence.

Clarified, as suggested.

— L296: omit "to a hemisphere; more specifically," as this explanation is not quite accurate.

Changed, as suggested.

— L340: Discussion section is section 4 (not 5)

Corrected.

— Figure captions: Perhaps specify what each row and column correspond to, as well as the colors of each line, to make it quicker to study each figure.

Following the Reviewer's request, we have added headings to the rows and columns, although they were already included in the sub-figure titles for easier identification of the content. Regarding lines – they were labelled with text info, and the labels also used the same colour scheme as the corresponding line.

---

## Author Response (AR2)

2025-03-31

**Response to the Editor's Review**

Dear Editor,

Thank you for your comments and suggestions. We have taken them into account in the manuscript. Below is our point-by-point response to your comments.

Regards,

on behalf of the authors
Andrzej Z. Kotarba

— 1) Reviewer 1: Question on "ground truth": I think, here it depends on what property you want to validate. There are some "ground truths" available e.g. in situ airborne observations if it is about physical cloud properties such as cloud particle number, size, shape.... In case of cloud classification, I see another limitation. That is the definition of the classification, which always differs depending on the type of observations you use. E.g., a human would identify DCC and cloud fraction different than a lidar. You may clarify in this section what type of quantity you refer to. A quick search showed some additional literature, which can be used here as references *[3 URLs, not shown here]*.

**Clarified** in the text. We fully agree. Our statement only referred to cloud type classification.

— 2) Reviewer 2: L357-363. Your additional study seems to be quite relevant. Why not including the full result, the table, in the manuscript?

The table has been **included** (as 'Table 6'), as suggested.

— 3) Tables: Table 1 and 2 show frequencies in % while Table 3 and 4 show normalized frequencies (I guess). This can be misleading when just looking at the Tables. I suggest to provide frequencies consistently with % or at least add "normalized frequencies" in the tables 3 and 4 where dimensionless frequencies are shown.

**Clarified in the text**. Tab. 3 and Tab. 4 refer to the frequency of clouds (also known as: number of clouds, fractional cloudiness, ratio of cloudy cases to all cases, etc.). The latter parameter is usually reported as a value between 0 to 1. It can be expressed as a value from 0% to 100% as well. However, using only the first notation <0,1> one can avoid an ambiguity associated with describing any relative changes in cloud frequency. Therefore, we prefer not to change the notation, thus **added an explanatory statement** in the table captions for clarity.